# Balancing Information Preservation and Computational Efficiency: L2 Normalization and Geodesic Distance in Manifold Learning

## Abstract

Distinguishable metric of similarity plays a fundamental role in unsupervised learning, particularly in manifold learning and high-dimensional data visualization tasks, by which differentiate between observations without labels. However, conventional metrics like Euclidean distance after L1-normalization may fail by losing distinguishable information when handling high-dimensional data, where the distance between different observations gradually converges to a shrinking interval. In this article, we discuss the influence of normalization by different p-norms and the defect of Euclidean distance. We discover that observation differences are better preserved when normalizing data by a higher p-norm and using geodesic distance rather than Euclidean distance as the similarity measurement. We further identify that L2-normalization onto the hypersphere is often sufficient in preserving delicate differences even in relatively high dimensional data while maintaining computational efficiency. Subsequently, we present HS-SNE (HyperSphere-SNE), a hypersphere-representation-system-based augmentation to t-SNE, which effectively addresses the intricacy of high-dimensional data visualization and similarity measurement. Our results show that this hypersphere representation system has improved resolution to identify more subtle differences in high-dimensional data, while balancing information preservation and computational efficiency.

## 1 Introduction

Similarity measurement is a fundamental problem in unsupervised learning, particularly in manifold learning and high-dimensional data visualization. Without labels, similarity measurements serve as the primary apparatus to establish the underlying latent structure between observations. Existing similarity metrics, like Euclidean distance, commonly rely on Euclidean space and $L^1$-normalization preprocessing, which preserve progressively less information as the dimension rises, thus falling victim of "the curse of dimensionality" (Köppen, 2000; Kiselev et al., 2019). In general, it is advised to avoid directly applying unsupervised learning with high-dimensional data. In tasks such as single-cell RNA sequencing analysis, high-dimensional cell-gene expression data are often visualized in 2-dimensional embeddings to explore the cell differentiation structure (Moon et al., 2019) or identifying new cell subtype.

It is common in a dataset that observations differ only in scale but not in feature compositions. This could be a result of technical effects or simply intrinsic data heterogeneity. A widely-adopted strategy is to first eliminate scale differences with normalization. This is especially critical when feature composition is the primary interest of investigation. Among all normalization schemes, $L^1$-normalization is perhaps the most prevalent so far, especially in Bioinformatics (Dillies et al., 2013; Townes et al., 2019), Information Retrieval (Qaiser & Ali, 2018) and Topic Modeling (Blei et al., 2003). In these tasks, each data point typically corresponds to a random observation sampled from a latent multinomial distribution. It is often overlooked, however, that $L^1$-normalization essentially projects each data point onto a simplex geometry, which gradually succumbs to the curse of dimensionality as the dimensionality rises.

The information of Euclidean distance after $L^1$-normalization unfortunately shrinks as the dimension increases. As the dimension rises further, observations begin to cramp to an increasingly small proximity on the manifold, losing the ability to distinguish observations (Köppen, 2000). Such loss of information negatively impact the downstream analysis. For instance, in t-SNE, a popular inter-point-distance based manifold learning algorithm (Van der Maaten & Hinton, 2008), as the dimensionality grows, distances between individual observations gradually gravitate towards an increasingly sharper Gaussian distribution, leading to their eventual portraiture as a single Gaussian cloud in the low-dimensional embedding.

Here, we question the necessity of using Euclidean distance after $L^1$-normalization as the primary distance metric and normalization scheme. In particular, we consider the plausibility of adopting $L^p$-normalization and geodesic distance as a replacement metric. Specifically, we propose projecting observations to a $\sum_i |\mathbf{x}_i|^p = C$ curvature instead of a $\sum_i \mathbf{x}_i = C$ simplex (with $C$ being a constant) and use the geodesic distance on the curvature as the similarity measurement. In addition, to balance performance and efficiency, we investigate whether the computationally friendly $L^2$-normalization, which projects observations to a hypersphere, is sufficient at information preservation in high-dimensional space while also leveraging the fact that its geodesic distance is simply the angular distance between two observations. Our contributions are the following.

- We systematically demonstrate and evaluates the capability of information preservation with $L^p$-normalization with regard to different dimensionalities. Specifically, we provide mathematical foundation that elucidates the futility of Euclidean distance after $L^1$-norm at discriminating data points of various degrees of affinity.

- We demonstrate that while being not the most information-captive normalization scheme, $L^2$-normalization and geodesic distance are often sufficient at preserving inter-observation differences, with diminishing returns with higher $p$ in $p$-norm.

- We introduce HS-SNE (**H**yper**S**phere-**SNE**), an augmented t-SNE visualization scheme that projects observations to a hypersphere and uses geodesic distances as the distance metric. HS-SNE achieves good performance in producing clearer boundaries between clusters in visualization while paying only a fraction of the computational cost in real single cell RNA-seq datasets, validating the effectiveness of $L^2$-normalization and geodesic distance system.

## 2 PROBLEM FORMULATION AND METHODS

In manifold learning, the principal objective is to learn a low-dimensional representation of the observations that retains sufficient information in demonstrating the relative similarity between observations in the high-dimensional space. Similarity measurement thus is at the center of manifold learning. Owing to the wide dynamic range of scale differences among observations due to technical factors, such as different total mRNA abundance between cells stemming from varying sequencing depths, data normalization has become an essential step in data preprocessing. Despite its importance, downstream effects of poorly-chosen normalization schemes to the learning outcome have, in our humble opinion, not been given sufficient attention. This is critical since normalization itself assumes an implicit similarity measurement framework as it in-effect projects the observations to a presumed manifold, where distance should be measured in a geodesic fashion accordingly (Li & Dunson, 2019). In an ideal case, post-normalization data points should preserve as much useful information as possible, as opposed to squeezing observations to a progressively cramped finite domain.

A straightforward and natural method for normalization is $L^1$-normalization, also known as total count normalization (Evans et al., 2018), a technique widely employed in the field of single-cell analytics (Hao et al., 2021; Wolf et al., 2018). The fundamental principle of this approach is to equalize the sum of all features between data points in order to erase technical differences among observations. Geometrically, $L^1$-normalization is equivalent to projecting $N$-dimensional observations to an $(N-1)$-dimensional hypersimplex, representing a hyperplane in an Euclidean space (Rispoli, 2008). $L^1$-normalization is a member of the more generic $L^p$-normalization family, defined as:

$$\mathbf{x}_{(i)}^{L^p \ normalized} = \frac{\mathbf{x}_{(i)}}{\sqrt[p]{\sum_i |\mathbf{x}_{(i)}|^p}},$$

with $p$ dictating the choice of the projection manifold. What is less investigated, is the information preservation capability of different $p$.

To elucidate the somewhat ambiguous concept of "information", we introduce the **I**nformation of **D**istance **D**istribution ($IDD$) metric for information preservation measurement. The derivation of $IDD$ is grounded in the concept of entropy from Information Theory. Given a distance distribution $Dist$, the $IDD$ is defined as follows:

$$IDD(Dist) = -\sum_{x=0}^{0.995} Pr(x) \log Pr(x),$$

where $Pr(x)$ denotes the distribution of the Min-max normalized inter-point distance. A larger $IDD$ indicates greater complexity and variability of the inter-point distances, hence a greater resolution and diversity in the post-normalization data. Conversely, a lower $IDD$ indicates a more concentrated inter-point distribution, where observations become equally distant and the sense of similarity differences is lost.

T-distributed stochastic neighbor embedding (t-SNE) (Van der Maaten & Hinton, 2008) and Uniform Manifold Approximation and Projection (UMAP) (McInnes et al., 2018) are two widely-used manifold learning visualization tools that learn from the pairwise distances within a dataset. They share a similar key idea: reconstruct the high-dimensional pairwise distance distribution in a low-dimensional space such that relative similarities between observations is preserved. Below is a brief introduction of t-SNE here, with UMAP follow suit.

Given a set of $N$ high dimensional $L^1$-normalized data points $\mathbf{x}_1, \dots, \mathbf{x}_N$ with $\mathbf{x}_i \in R^n$, it is assumed that the distance distribution follows a Gaussian distribution and the conditional probability of $\mathbf{x}_i$ relative to $\mathbf{x}_j$ ($i \neq j$) can be computed as:

$$p_{j|i} = \frac{\exp\left(-\|\mathbf{x}_i - \mathbf{x}_j\|^2/2\sigma_i^2\right)}{\sum_{k \neq i} \exp\left(-\|\mathbf{x}_i - \mathbf{x}_k\|^2/2\sigma_i^2\right)}. \tag{1}$$

In this process, Euclidean distance is used for the similarity measurement $\|\mathbf{x}_i - \mathbf{x}_j\|$ in the high-dimensional space. $\sigma_i$ is a hyper-parameter that regulates the item-wise Gaussian kernel standard deviation width. To create a symmetrical distance between $i$ and $j$, $p_{j|i}$ and $p_{i|j}$ are merged into $p_{ij}$ via

$$p_{ij} = \frac{p_{j|i} + p_{i|j}}{2N},$$

with $\sum_{i,j} p_{ij} = 1$. For the embedding space of $d$ dimensions ($d << N$), let $\mathbf{y}_1, \dots, \mathbf{y}_N \in R^d$ denote the corresponding low-dimensional embedding. t-SNE learns a mapping between $\mathbf{x}$ and $\mathbf{y}$ such that the inter-point distances between $\mathbf{x}$ and $\mathbf{y}$ are aligned from an information perspective. Specifically, it forces the inter-point distances in the $d$-dimensional space to follow item-wise t-distributions. The information distance $q_{ij}$ between $i$ and $j$ is then computed as follows:

$$q_{ij} = \frac{\left(1 + \|\mathbf{y}_i - \mathbf{y}_j\|^2\right)^{-1}}{\sum_k \sum_{l \neq k} \left(1 + \|\mathbf{y}_k - \mathbf{y}_l\|^2\right)^{-1}}$$

Ultimately, t-SNE employs Kullback-Leibler divergence (KL divergence) as a measurement of discrepancies between the two distributions in the high-dimensional and low-dimensional spaces and set off to minimize the KL divergence through gradient decent.

$$\text{KL}(P\|Q) = \sum_{i \neq j} p_{ij} \log \frac{p_{ij}}{q_{ij}}$$

UMAP essentially follows the same philosophy but slightly deviates from t-SNE in distribution and loss designs.

While much focus has been spent on distribution and loss designs, it is often overlooked that a poor choice of normalization scheme could devastate the learning outcome. Employing a normalization scheme with a low $IDD$ blurs the differences between observations, often leading to a false perception of data points being randomly dispersed in a wide dynamic range while overriding any of the inherent structures between observations, such as clustering behaviors, as depicted in Figure 1.

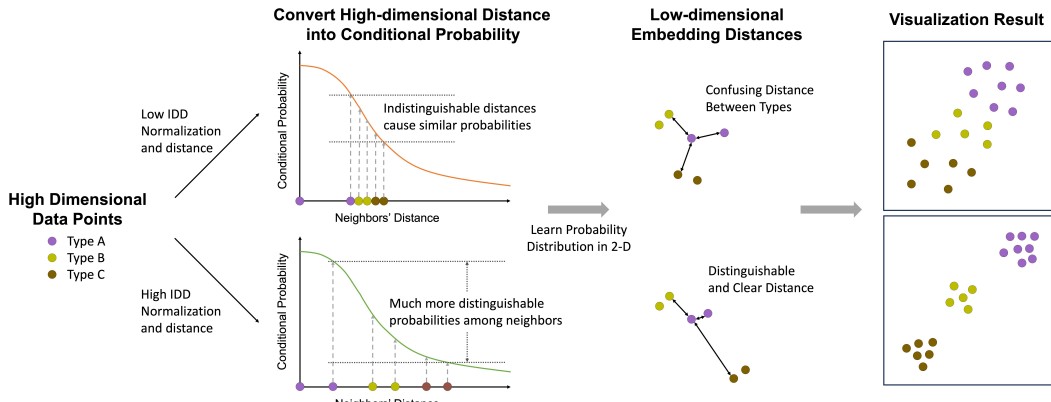

Figure 1: Low $IDD$ implies that the pairwise distances between high-dimensional points are nearly identical, leaving no effective patterns for manifold learning methods to learn. This results in ambiguous and seemingly random visualizations in low-dimensional embeddings.

$L^p$-normalization with higher $p$ values can map points more uniformly, thereby intuitively ensuring that the distances do not fall into a very narrow interval, as illustrated in Figure 2a. However, for $p > 2$, the geodesic distance does not have a closed-form solution according to existing literature (Kiryati & Székely, 1993; Davis et al., 2017; Li & Dunson, 2019; Fontenot et al., 2022). In the case of $L^1$-normalization, the geodesic distance coincides with the Euclidean distance. For $L^2$-normalization, the manifold constitutes a hypersphere; hence, the geodesic distance is the shortest arc-length between two points, also referred to as the great circle distance or angular distance, which can be represented as:

$$\|\mathbf{x}_i - \mathbf{x}_j\|_{geo} = \arccos\left(\mathbf{x}_i \cdot \mathbf{x}_j\right).$$

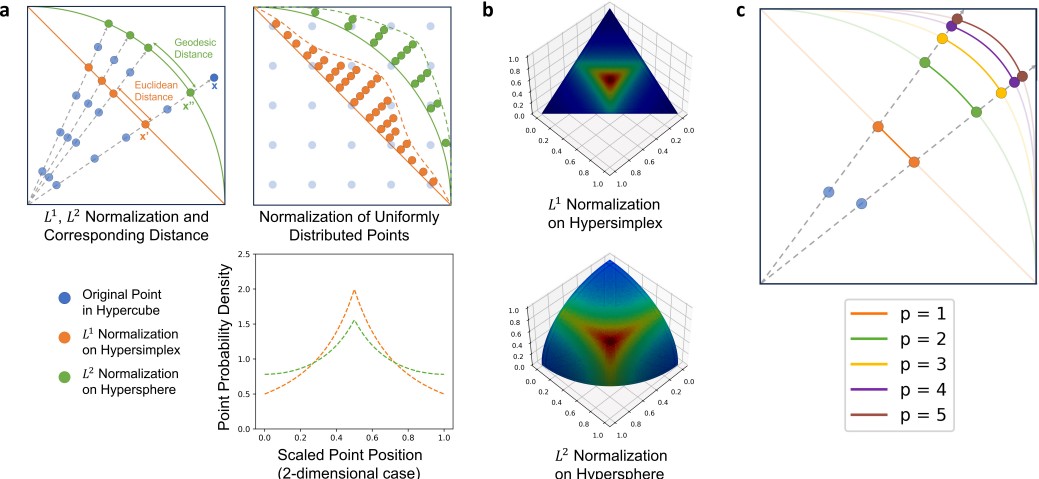

Figure 2: **a.** Illustration of $L^1$ and $L^2$ normalization in the 2-dimensional case. The distribution of $L^1$-normalized points is denser than that of $L^2$-normalized points, forming a sharp peak. **b.** Heatmap of 3-dimensional case point probability density. **c.** Illustration of $L^p$-normalization and the geodesic distance between two normalized points in 2-dimensional space for varying values of $p$.

In 2-dimensional case, since the manifold of the $L^p$-normalized points is just $\mathbf{x}_{(1)}^p + \mathbf{x}_{(2)}^p = 1$, where $x_{(i)}$ represents the $i_{th}$ feature of data point $x$, like what is shown in Figure 2c, we can still use line integral to calculate the geodesic distance between $\mathbf{x}$ and $\mathbf{y}$ (assuming that $\mathbf{y}_{(1)} \geq \mathbf{x}_{(1)}$ in

closed-form,

$$\int_{\mathbf{x}_{(1)}}^{\mathbf{y}_{(1)}} \sqrt{1 + (\frac{dy}{dx})^2} dx$$

where $\frac{dy}{dx} = -x^{p-1}(1 - x^p)^{1/p-1}$ for $p$-norm normalization.

To investigate the geodesic distance in $L^p$-normalization for $p > 2$ and more than 2-dimensional, we introduce an estimation method, inspired from Isomap (Tenenbaum et al., 2000). This method leverages the approximation of Euclidean distance to any distances within a localized structure. Initially, we randomly sample $n$ points $\mathbf{m}_1, \mathbf{m}_2, \ldots, \mathbf{m}_n$ on the $L^p$-normalization manifold and subsequently construct a k-NN graph $G$ for these points. The geodesic distance can then be estimated as the sum of the shortest path on the k-NN graph between the two sampling points $\mathbf{m}_i, \mathbf{m}_j$ nearest to the target points $\mathbf{x}_i, \mathbf{x}_j$, plus the double-ended Euclidean distances, as expressed in the equation:

$$\|\mathbf{x}_i - \mathbf{x}_j\|_{geo} = \|\mathbf{x}_i - \mathbf{m}_i\| + d(G, \mathbf{m}_i, \mathbf{m}_j) + \|\mathbf{x}_j - \mathbf{m}_j\|.$$

However, the accuracy of this estimation greatly depends on the sample size $n$ and the dimensionality $D$. Employing Dijkstra's algorithm, the time complexity for finding the shortest path between two sampling points is $O(n^2)$, and $n \propto c^D$ to maintain a consistent sampling density $c$ across varying dimensionalities. Consequently, this complexity can escalate rapidly, rendering the method intractable for high-dimensional data.

Moreover, the loss of information becomes increasingly significant with the rise in dimensionality due to the "curse of dimensionality" (Köppen, 2000), as depicted in Figure 3a, where we uniformly generated $100,000$ pairs of points in a $D$-dimensional hypercube space to show the distance distribution change versus dimensionality increasing. Figure 3b clearly show the $IDD$ change versus dimensionality increasing, the decline of $IDD$ for geodesic distance after $L^2$ normalization is much slower than Euclidean distance after $L^1$ normalization. Not only uniformly distributed point pairs, the point pairs sampled from a normal distribution and Dirichlet-multinomial (DirMult) distribution (Sun et al., 2018) also validated this property, which are common in real data.

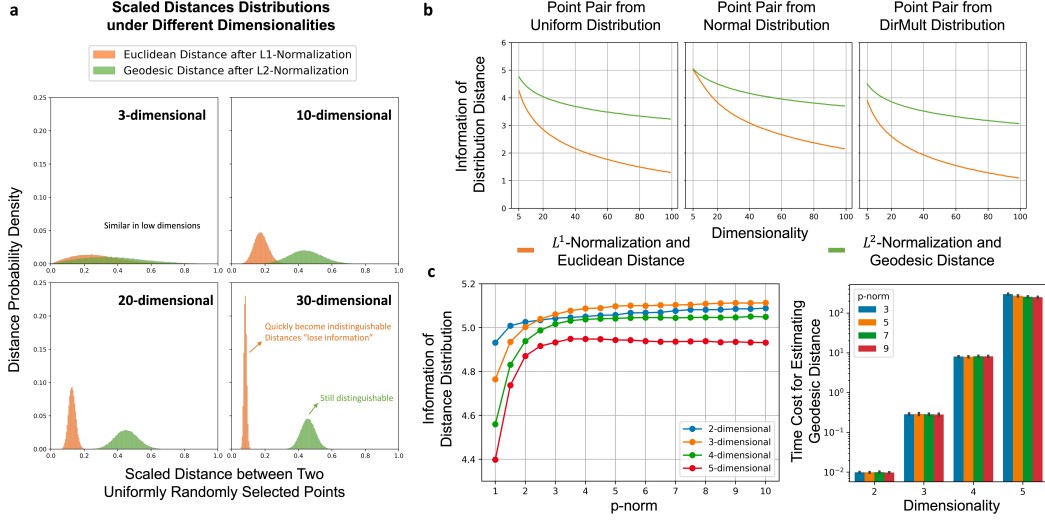

Figure 3: **a.** As dimensionality increases, Euclidean distances under $L^1$-normalization quickly converge to a small interval, forming a sharp peak, whereas those under $L^2$-normalization and geodesic metrics remain distinguishable. **b.** For points sampled from different distributions, $L^2$-normalization consistently outperforms $L^1$-normalization in preserving $IDD$. **c.** $L^2$-normalization strikes an effective balance between enhancing $IDD$ and maintaining computability, proving to be sufficiently good compared to other $p$-norms.

From the perspective of enhancing $IDD$, the marginal benefit of increasing $p$ diminishes rapidly as shown in Figure 3c. The setting of $p = 2$ already achieves substantial improvement compared to $p = 1$. Given that the geodesic distance in $L^2$-normalization has a closed-form solution, resulting in $O(1)$

time complexity, $L^2$-normalization strikes an effective balance between information preservation and computational efficiency.

The manifold formed by $L^2$-normalization is a hypersphere, which introduces another challenge when applying it to t-SNE. In t-SNE, distances are transformed into conditional probabilities using a Gaussian distribution kernel (Van der Maaten & Hinton, 2008); however, this distribution is only appropriate for modeling random distances in Euclidean space. In a spherical space, the appropriate distribution for modeling arc-length, while retaining Normal properties, is the von-Mises distribution, given by:

$$f(x; \mu, \kappa) = \frac{1}{2\pi I_0(\kappa)} \exp\left(\kappa \cos(x - \mu)\right),$$

where $I_0(\kappa)$ is the modified Bessel function making the distribution sum to 1, $\mu$ is the mean direction and $\kappa$ is the concentration parameter (Watson, 1982). We can get a kernel function derived from the von-Mises distribution, which is

$$\exp\left(\kappa \cos(\|\mathbf{x}_i - \mathbf{x}_j\|_{geo}) - \kappa\right).$$

We add a "$-\kappa$" term in the exponential function to make sure that the kernel returns 1 when $\mathbf{x}_i$ is equal to $\mathbf{x}_j$ exactly. Therefore, the conditional probability of $\mathbf{x}_i$ to $\mathbf{x}_j$ is

$$p_{j|i} = \frac{\exp\left(\kappa \cos(\|\mathbf{x}_i - \mathbf{x}_j\|_{geo}) - \kappa\right)}{\sum_{k \neq i} \exp\left(\kappa \cos(\|\mathbf{x}_i - \mathbf{x}_j\|_{geo}) - \kappa\right)}$$

We propose replacing the Gaussian distribution kernel in Equation 1 with this von-Mises distribution kernel when using t-SNE with $L^2$-normalization and geodesic distance. We refer to this adaptation as HS-SNE, serving as a validation for our $L^2$ Hypersphere normalization and distance metric system.

## 3 EXPERIMENTAL RESULTS

To assess the efficacy of our methods, we will conduct simulation experiments with known ground truth labels. We will test clustering as a downstream task on low-dimensional embeddings obtained from both t-SNE and HS-SNE. Subsequently, we will employ real single-cell RNA sequencing data, which is inherently high-dimensional and intricate, as a testing ground for our visualization techniques.

To further validate our approach, a k-NN graph on the low-dimensional embeddings will be constructed, and the accuracy with which neighbors are classified as belonging to the same cluster will serve as a performance metric. This will offer insight into the granularity and correctness of the cluster formations yielded by our methodology.

In addition to this, we will explore the application of the $L^2$-normalization hypersphere system to UMAP, aiming to demonstrate its potential to enhance all manifold learning algorithms based on distance relationships. Our endeavor is to validate that $L^2$-normalization method is not only a substantial improvement for t-SNE but also a useful tool that can be adapted to a variety of manifold learning techniques, thereby broadening its applicability and impact in the field.

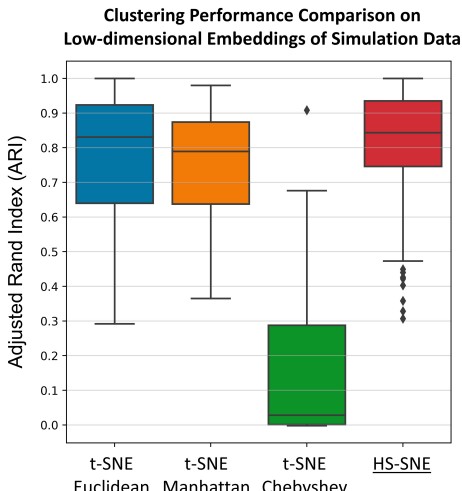

Figure 4: HS-SNE improves clustering performance on low-dimensional embeddings compared t-SNE with other distance metrics.

### 3.1 HS-SNE VISUALIZATION IMPROVES CLUSTERING ON SIMULATION DATA

Clustering is a fundamental downstream task closely associated with dimensionality reduction. To address the curse of dimensionality, clustering is typically performed on lower-dimensional em-

beddings. In this study, we generated labeled high-dimensional data and utilized t-SNE with $L^1$-normalization and various commonly used distance metrics, and HS-SNE, to visualize the data in a 2-dimensional space. Subsequently, we used K-Means clustering to assess the differences in performance.

We employed the Dirichlet-multinomial Mixture Model for data generation, a well-established model for fitting single-cell sequencing biological data (Sun et al., 2018). Using this model, we created data in $2,000$ dimensions, consisting of three distinct types, with each type consisting of $300$ instances. During the data generation process, 25 features between each pair of types were subjected to varying parameters $\alpha$ within the model to introduce diversity. The expected result was to ensure the preservation of variance between data points of different types in the lower-dimensional embedding, thereby yielding distinguishable clustering results.

Subsequently, K-Means clustering was applied to each of the derived low-dimensional embeddings, with the number of clusters set to 3. To quantify the clustering performance, we employed the Adjusted Rand Index (ARI). After conducting the experiment across 100 iterations, the results in Figure 4 indicate that HS-SNE surpasses t-SNE, regardless of whether the Euclidean, Manhattan, or Chebyshev distance metric is applied within t-SNE.

### 3.2 HS-SNE Produce Clearer Visualization and More Accurate Embedding k-NN Graph in Real Biological Data

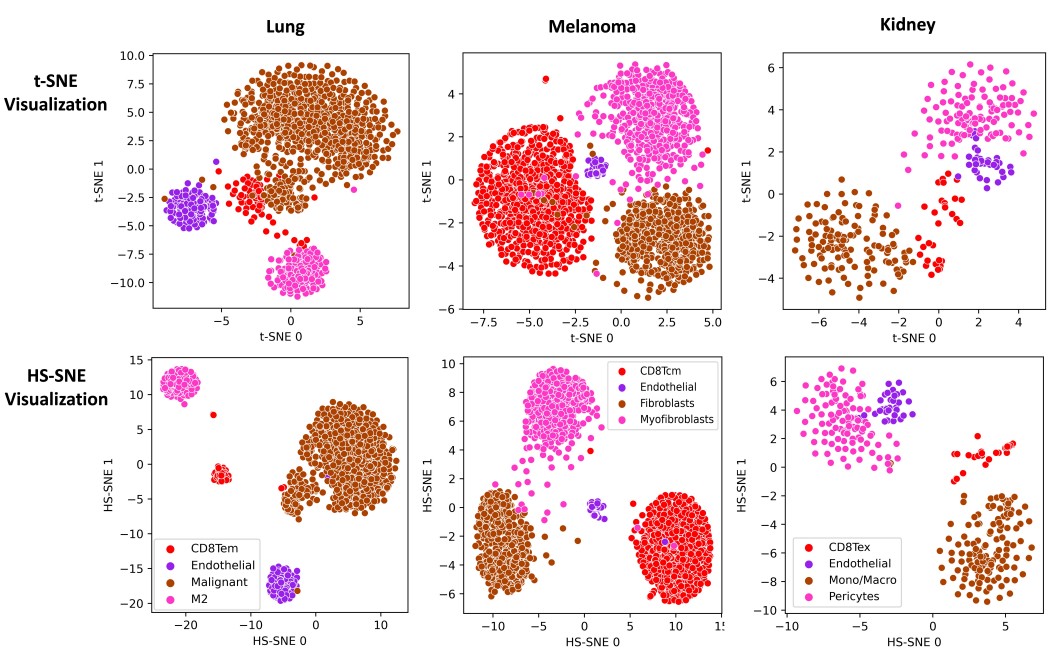

Figure 5: HS-SNE shows better 2-dimensional visualization results with clearer boundaries between clusters than t-SNE due to the informative $L^2$-normalization and geodesic distance.

Single-cell RNA sequencing is a popular field where biologists need to visualize high-dimensional cell-gene expression data to 2-d embeddings to identify cell types (Cakir et al., 2020). We utilized real biological datasets related to the tumor micro-environment, derived from single-cell RNA sequencing studies on lung cells (Guo et al., 2020), melanoma cells (Li et al., 2020), and kidney cells (Zhang et al., 2021), to evaluate HS-SNE. These datasets were obtained from the TISCH project (Sun et al., 2021; Han et al., 2023), and we select some biological meaningful cell types in each dataset to demonstrate the results. Figure 5 illustrates the comparison between HS-SNE and t-SNE, depicting that the clusters in HS-SNE are more concentrated as anticipated, which is vital for identifying cell types and subtypes.

The scaled data points pairwise distance distribution derived from the lung cells dataset in Figure 6 substantiates our assertion that $L^2$-normalization on the hypersphere coupled with geodesic distance can yield more informative distances. The distances from HS-SNE have two clear peaks indicating inter-cell type and intra-cell type distances, thus leading to visualization with clearer boundaries between clusters. The Euclidean distance in t-SNE, when accompanied by $L^1$-normalization on the hypersimplex, falls into a very small interval, manifesting as only one sharp peak.

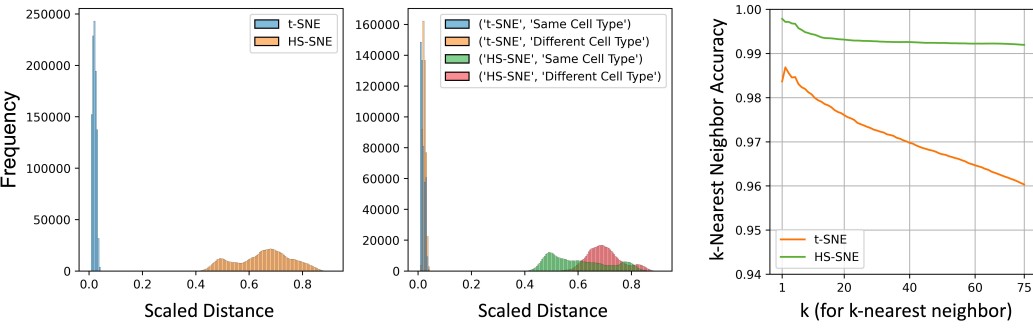

Figure 6: Scaled distance distribution in real lung cells scRNA-seq dataset shows that HS-SNE can better preserve distinguishable information than t-SNE. The k-Nearest Neighbor Accuracy on low-dimensional embeddings also shows that HS-SNE outperform t-SNE visualization in terms of local structure preservation.

In methods grounded on SNE, a crucial step involves the construction of the k-Nearest Neighbor (k-NN) graph. The precision with which the nearest neighbors of a data point are classified as belonging to the same type serves as a indicator of the quality of the constructed k-NN graph. To evaluate this, we computed the accuracy of the k-Nearest Neighbors in the 2-dimensional embeddings of the lung cells dataset, with $k$ ranging from 1 to 75, considering the smallest category in this dataset comprises 75 examples. The results in Figure 6 illustrate that HS-SNE consistently outperforms t-SNE in k-nearest neighbor accuracy. This suggests that, in comparison to traditional t-SNE employing Euclidean distance, HS-SNE holds the capacity to generate a more accurate k-NN graph in low-dimensional embedding, thereby enhancing the reliability of downstream analysis.

### 3.3 L2-normalization and Geodesic Distance also benefits UMAP

Uniform Manifold Approximation and Projection (UMAP) is another dimensionality reduction technique that is particularly effective for visualizing high-dimensional data in lower-dimensional space, with better ability to preserve both local and global structures of the data. Like t-SNE, UMAP is also based on distance measures, so that $L^2$-normalization and geodesic distances can be employed to enhance the granularity of the resulting projections. We applied UMAP and a variant, Hypersphere UMAP, which integrates $L^2$-normalization and geodesic distances, to visualize the lung cells data above. The results demonstrate a significant improvement in the preserva-

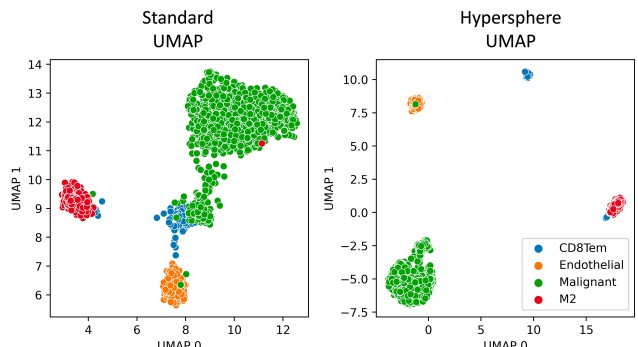

Figure 7: $L^2$-normalization and geodesic distance are also applicable for improving other distance-based manifold learning such as UMAP as shown in the lung cells dataset.

tion of clusters structure of lung cell clusters, thereby underscoring the potential of $L^2$-normalization and geodesic distances in unraveling intricate data relationships in the high-dimensional dataset.

## 4 DISCUSSION

In this paper, our primary focus has been on the simple $L^p$-normalization. We propose that $L^p$-normalization is general with extensive application across various fields. Given the widespread usage of $L^1$-normalization, we advocate for the adoption of the similarly structured $L^2$-normalization for manifold learning in high-dimensional data. This alternative approach not only enhances performance but also requires minimal alterations to the existing model. Furthermore, the underlying methodology of geodesic distance in $L^2$-normalization resembles angular distance or cosine distance, providing a theoretical foundation for choosing these measures over the Euclidean distance. In all, this report provides the theoretical underpinnings of these alternative distance metrics, reinforcing their utility in analyzing high-dimensional data by manifold learning.

## 5 CONCLUSION

In conclusion, we explored the impact of various $L^p$-normalization and geodesic distance on manifold learning visualization. We showed that higher $L^p$-normalization can preserve more distinguishable distance information. However, estimating the geodesic distance for $p > 2$, which lacks a closed-form solution, incurs a significantly large time complexity, rendering it impractical for high-dimensional data. We illustrated that the marginal benefits of information preservation diminish as $p$ increases. Through extensive experiments, $L^2$-normalization exhibited commendable capability in preserving information, while maintaining a balance with computational efficiency. To validate the advantages of the $L^2$-normalization hypersphere system compared to t-SNE with $L^1$-normalization and Euclidean distance, we developed HS-SNE, which produced clearer visualization results in both simulated datasets and real biological datasets, and improving the downstream clustering tasks, helping identifying important cell types. Consequently, we advocate for the utilization of $L^2$-normalization and geodesic distance in manifold learning to mitigate scale differences and preserve the distinguishable distance information inherent in high-dimensional data.

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

# A   APPENDIX

## A.1   ILLUSTRATIVE EXAMPLE SETTINGS IN FIGURE 2

The illustrative scenario involving a random $n$-dimensional observation, denoted as $\mathbf{x} \in \mathbf{R}^n$, comprising $n$ features, with $\mathbf{x}_{(i)}$ representing the $i^{th}$ feature. Assume that each feature follows a uniform distribution ranging from 0 to 1, satisfying $\forall i,\ 0 \leq \mathbf{x}_{(i)} \leq 1$. In this setting, the sample space of $\mathbf{x}$, labeled as $\mathbf{S}$, corresponds to an $n$-dimensional hypercube. For instance, if $n = 2$, then $\mathbf{S}$ would be a 2-dimensional hypercube, essentially forming a square.

Denote the sample spaces of $L^p$-normalized points as $\mathbf{S}_{Lp}$, and let $PDF(d)$ be the probability density of normalized points positioned at distance $d$ from the point $(0, 1)$. $PDF_{\mathbf{S}_{L1}}$ and $PDF_{\mathbf{S}_{L1}}$ are as follows.

$$PDF_{\mathbf{S}_{L1}}(d) = \begin{cases} \frac{1}{\sqrt{2}(\sqrt{2}-d)^2}, & d \in [0, \frac{\sqrt{2}}{2}) \\ \frac{1}{\sqrt{2}d^2}, & d \in [\frac{\sqrt{2}}{2}, \sqrt{2}] \end{cases} \quad , PDF_{\mathbf{S}_{L2}}(d) = \begin{cases} \frac{1}{2\cos^2(d)}, & d \in [0, \frac{\pi}{4}) \\ \frac{1}{2\cos^2(\frac{\pi}{2}-d)}, & d \in [\frac{\pi}{4}, \frac{\pi}{2}) \end{cases} \quad .$$

We scale all $PDF$ to align the domain with $x \in [0, 1]$. This scaling process is denoted as

$$Pr(x) = \mu \cdot PDF(\mu \cdot x),$$

where $\mu = \max(d)$. The graph of $Pr(x)$ is depicted in Figure 2a, revealing that the probability density of $L^2$-normalized points on sphere appears smoother, while the $L^1$-normalized points exhibits a steeper shape akin to a sharper peak. Furthermore, we present a heatmap showing normalized points density in a 3-dimensional setting in Figure 2b. Importantly, this distinction in sharpness becomes increasingly pronounced as dimensionality grows.

