# OpenReview forum: "Balancing Information Preservation and Computational Efficiency: L2 Normalization and Geodesic Distance in Manifold Learning"
_ICLR.cc/2024/Conference — Submitted to ICLR 2024_

### Official Review · Reviewer_mtLk · 2023-10-26

**Soundness:** 2 fair
**Presentation:** 3 good
**Contribution:** 1 poor
**Rating:** 3
**Confidence:** 5

**Summary:**

In their paper "Balancing information presevration and computational efficiency: L2 normalization and geodesic distance in manifold learning", the authors suggest a t-SNE modification called HS-SNE that does two things: (1) uses L2 normalization of the input data to project it to the hypersphere; (2) replaces Gaussian distribution with von Mises distribution to compute high-dimensional pairwise affinities between points. Using simulated as well as real biological data, the authors argue that their approach outperforms L1 normalization and standard t-SNE commonly used to analyze single-cell RNA-seq data.

**Strengths:**

I am interested in manifold learning and scRNA-seq data, and think that exploring the effect of data normalization is an interesting and under-studied topic. I think the paper is on-topic, suggests a somewhat novel approach (HS-SNE), and uses quantitative evaluation.

**Weaknesses:**

That said, I also think that the paper does not convincingly show benefits of HS-SNE, does not present the necessary ablation experiments, and does not show all the relevant controls. In fact, the modification (2) does not do anything at all, because the way authors defined it, von Mises distribution is equivalent to Gaussian. I am afraid the paper does not rise to the ICLR level of general interest.

**Questions:**

MAJOR COMMENTS

* Once the points are projected onto the hypersphere and have length 1, cosine of the geodesic distance is simply cosine similarity, which is 1 minus one half of the squared Euclidean distance. So the expression exp(kappa * cos(geo_dist) - kappa) used in section 2 to define p_{ij} values in t-SNE is exactly equivalent to exp(-eucl_dist^2 / (2/kappa)), which is *exactly* the expression used by standard t-SNE. If kappa is tuned like sigma is tuned in t-SNE (to achieve a given perplexity), this is an absolutely identical procedure and will make no difference.

* In the experiment in Section 3.1, HS-SNE is compared to t-SNE after L1 normalization. What is crucially missing, is standard t-SNE after L2 normalization. As explained above, it must be equivalent to HS-SNE.

* As a sanity check, I would also like to see t-SNE of the raw data (without any normalization) in Figure 4. On the other hand, t-SNE with Manhattan and with Chebyshev distances are not really needed there, in my opinion.

* In Figure 4, I don't see any difference between t-SNE(L1) and HS-SNE: blue vs red. Why do the authors conclude that HS-SNE "surpasses" t-SNE?? I am really confused by that intepretation.

* Figure 5 and Figure 6c are more convincing, and do show a noticeable difference between t-SNE(L1) and HS-SNE. Again, what is missing here is a comparison to t-SNE(L2), which should be equivalent to HS-SNE. Then the message of the paper would reduce to "For scRNA-seq data, L2 normalization is better than L1 normalization", which may be interesting in itself, but probably better suited for a bioinformatics journal.

* kNN accuracy used in Figure 6 is a good metric, but why use kNN accuracy in 2D after t-SNE, instead of using kNN accuracy directly in the high-dimensional space? That would be a more direct measure of the normalization quality.


MINOR COMMENTS

* IDD formula on page 3 -- why does the sum go until 0.995?

* Section 3.2: standard practice in scRNA-seq literature is to apply PCA first, keeping 50-100 dimensions, before doing t-SNE. It would be interesting to know if this alleviates the problem of L1 normalization or not. I.e. it would be interesting to add t-SNE(PCA(L1)) pipeline to the comparison.

---

### Official Review · Reviewer_FzWv · 2023-10-30

**Soundness:** 2 fair
**Presentation:** 3 good
**Contribution:** 2 fair
**Rating:** 5
**Confidence:** 4

**Summary:**

The primary objective of this paper is to refine the normalization process employed in existing unsupervised learning algorithms. In a more detailed sense, the paper presents the novel idea of utilizing distance distribution and its related information to measure information loss, thereby indicating that earlier methods might lead to unidentifiable distances. To mitigate this, the paper recommends replacing the Gaussian distribution kernel with a Von Mises distribution kernel when implementing the t-SNE method.

**Strengths:**

1.As the dimensionality increases, the tendency of previous methods to result in indistinguishable distance distributions becomes more pronounced.

2.Empirical experiments demonstrate the effectiveness of the proposed method in comparison to the original t-SNE.

**Weaknesses:**

1.I question whether the information from distance distribution could serve as an effective measure of information loss, given the problem background.

2.The dataset used in section 3.2 needs a more detailed introduction. The phrase "some biological meaningful" could potentially lead to ambiguity.

**Questions:**

Why is the information from distance distribution effective in representing the information? Wouldn't metrics considering inter or intra-class distances, which account for the "correct distance for clustering," be a more appropriate choice for representing distance information? Is it possible to have a highly distinguishable distance distribution but still end up with significantly incorrect clustering results?

---

### Official Review · Reviewer_2wH8 · 2023-11-07

**Soundness:** 2 fair
**Presentation:** 2 fair
**Contribution:** 2 fair
**Rating:** 3
**Confidence:** 5

**Summary:**

1.	The paper establishes a robust mathematical foundation, highlighting the limitations of using Euclidean distance after the L1-norm for distinguishing data points with varying degrees of affinity.
2.	While not the most information-captive method, the paper demonstrates the effectiveness of L2-normalization and geodesic distance in preserving inter-observation differences, especially with lower p values in p-norm.
3.	The introduction of HS-SNE (HyperSphere-SNE) presents an innovative approach to data visualization. By projecting observations onto a hypersphere and employing geodesic distances, it significantly improves cluster boundaries in visualizations, all while reducing computational costs in single-cell RNA-seq datasets.

**Strengths:**

1.	The paper establishes a robust mathematical foundation, highlighting the limitations of using Euclidean distance after the L1-norm for distinguishing data points with varying degrees of affinity.
2.	While not the most information-captive method, the paper demonstrates the effectiveness of L2-normalization and geodesic distance in preserving inter-observation differences, especially with lower p values in p-norm.
3.	The introduction of HS-SNE (HyperSphere-SNE) presents an innovative approach to data visualization. By projecting observations onto a hypersphere and employing geodesic distances, it significantly improves cluster boundaries in visualizations, all while reducing computational costs in single-cell RNA-seq datasets.

**Weaknesses:**

1.	The contribution of the paper is limited, considering that the authors simply replace the original L1-normalization term with an L2-normalization term to solve the dimensional catastrophe problem, with no innovative paradigm contribution presented. Moreover, this approach can only alleviate the problem to a certain extent without a milestone breakthrough.
2.	As claimed in the article INTRODUCTION, the advantage of L2-normalization is maintaining sufficient information in high-dimensional spaces, which lacks subsequent real datasets’ experimental validation. In addition, the dataset used lacked description in the paper, and I had difficulty confirming that the authors used a high-dimensional real dataset.
3.	The experimental part is not sufficient to verify the validity of the method. As far as the experiment in Fig. 5 is concerned, there are too few comparison methods and only one classical t-SNE algorithm, which is not convincing. Moreover, more models are expected for the experiment in Fig. 7 to demonstrate the effectiveness of the L2 -normalization term.

**Questions:**

1.	The contribution of the paper is limited, considering that the authors simply replace the original L1-normalization term with an L2-normalization term to solve the dimensional catastrophe problem, with no innovative paradigm contribution presented. Moreover, this approach can only alleviate the problem to a certain extent without a milestone breakthrough.
2.	As claimed in the article INTRODUCTION, the advantage of L2-normalization is maintaining sufficient information in high-dimensional spaces, which lacks subsequent real datasets’ experimental validation. In addition, the dataset used lacked description in the paper, and I had difficulty confirming that the authors used a high-dimensional real dataset.
3.	The experimental part is not sufficient to verify the validity of the method. As far as the experiment in Fig. 5 is concerned, there are too few comparison methods and only one classical t-SNE algorithm, which is not convincing. Moreover, more models are expected for the experiment in Fig. 7 to demonstrate the effectiveness of the L2 -normalization term.

---

### Meta-Review · Area_Chair_kXT8 · 2023-12-05

**Metareview:**

The paper proposes to employ L2 normalization of the input data along with the geodesic distance on the hypersphere for high-dimensional data. The authors then propose a modification of the t-SNE algorithm called HS-SNE using this method, where the Gaussian distribution is replaced with the von Mises distribution.

Reviewers generally agree that the proposed idea is novel and that the HS-SNE improves cluster boundaries in visualization compared with t-SNE. However, they also raise many concerns, including

- The overall contribution is limited in novelty (Reviewer 2wH8)

- The experimental validation needs to be stronger. It's not clear if the real data sets used are high-dimensional (Reviewer 2wH8);
Fig.4 does not show that  HS-SNE  surpasses t-SNE, as claimed in the paper (Reviewer mtLk)

- I also agree with Reviewer mtLk that the definition of the IDD, with the upper limit being 0.995, is not justified or motivated.

The authors did not provide a rebuttal.

**Justification For Why Not Higher Score:**

Reviewers are unanimous in their assessment of the paper.

**Justification For Why Not Lower Score:**

N/A

---

### Decision · Program_Chairs · 2024-01-16

Reject